# Analysis Projection of the Fulfillment of Priority Facilities and Infrastructures for Vocational High School/Sekolah Menengah Kejuruan (SMK) Using System Dynamic to Increase School Participation Rates in Central Kalimantan Province, Indonesia

**Hamiduddin Arief Kaenong** * 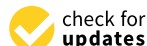, **Mohammad Benny Alexandri** and **Yogi Suprayogi Sugandi**

Fakultas Ilmu Sosial Dan Ilmu Politik, Universitas Padjadjaran, Jalan Dago Bukit Utara No. 25, Bandung 40135, Provinsi Jawa Barat, Indonesia; mohammad.benny@unpad.ac.id (M.B.A.); yogi.suprayogi@unpad.ac.id (Y.S.S.)
* Correspondence: hamiduddin21001@mail.unpad.ac.id; Tel.: +62-81355662688

**Abstract:** The fulfillment of facilities and infrastructures is the obligation of the provincial government, including the provincial government of Central Kalimantan. This study aims to find out whether the provincial government of Central Kalimantan can meet SMK priority facilities and infrastructures and to know the year of achievement. The research uses a quantitative method in the form of a dynamic system with the help of the vensim PLE64 application version 9.2.0.0 using profile data for each SMK and the Budget Execution List/Daftar Pelaksanaan Anggaran (DPA) of the education office from 2017 to 2022. The results show that if the budget for priority facilities and infrastructures is maintained, then in 2028, priority facilities and infrastructures will equal the ideal number. Furthermore, if the Central Kalimantan provincial government runs a scenario of increasing the growth of the SMK facility and infrastructure budget, uses the budget allocation for SMK facilities and infrastructures for only priority facilities and infrastructures, or uses the budget allocation for SMK facilities and infrastructures for SMKs first, then the fulfillment of facilities and infrastructures will be achieved 2–3 years earlier.

**Keywords:** vocational high school; government budget; school facilities and infrastructures; school participation; system dynamic

## 1. Introduction

The problem of educational participation is still a problem today. This is related to equal opportunities to continue secondary education. The 1945 Constitution affirms that every citizen has the right to education [1]. The government wants to push the school participation rate (SPR) among the high school-aged population as high as possible or minimize the dropout rate. This is in line with the availability of a sufficient number of high school study groups for students (SMK 1:25, where the maximum is 1:36), and there are more study groups than classrooms (1:0.9) [2]. Based on data from the national net participation rate (NPR) and the national gross participation rate (GPR), the national NPR is 68.90%, and the GPR is 97.52% (Kemdikbudristek RI, 2022). In addition, the government has made the education sector the largest allocation of state budget/anggaran pendapatan dan belanja negara (APBN) funding, namely as much as 20% or IDR 550 trillion, the largest in several ministries and institutions (as much as 45.63%), and transfers to the regions are as much as 54.37% [2].

Regional education balance/neraca pendidikan daerah (NPD) data for 2021 also show that the NPR and GPR for secondary education in Central Kalimantan Province are 60.26% and 88.60%, respectively, and the number of SMK and senior high school/sekolah menengah atas (SMA) students is 36,817 students and 60,634 students [2]. The Central



Kalimantan NPR and GPR Prov are still under the national NPR and GPR. This can be seen that, at school age (16–18 years) in secondary education, only 60.26% of Central Kalimantan's population attends school, the remaining 28.34% are residents outside the age of 16–18 years or residents outside Central Kalimantan Province. The NPR and GPR for secondary education in 2021 are compared to the NPR and GPR for junior high school in 2018 because those in secondary education in 2021 were in 2018 junior high school. The NPR and GPR for junior high school/equivalent in Central Kalimantan in 2018 were 73.5% and 100.3%, respectively, and the number of junior high school students is 108,822 students [3]. If you look at the difference in the NPR and GPR for secondary and junior high schools, this shows that not all residents of Kalimantan are continuing their education from basic education (SMP (Sekolah Menengah Pertama/Junior High School)) to secondary education (SMA/SMK). There are still junior high school-aged residents who do not continue to secondary education, namely around 13.24% based on the NPR and 11.70% based on the GPR, the number of students as many as 11,371 (number of junior high school students (108,822 people) minus the number of high school students (60,634 people) + SMK (36,817 people) [2].

Before 2017, management of vocational schools in Central Kalimantan Province was carried out by the regency/city government. The management of vocational schools by the Central Kalimantan provincial government includes facility and infrastructure management based on law no. 23 of 2014. Based on this law, the central government has the authority to set school management standards, and the regency/city government is mandated to manage elementary and junior high schools [4]. The number of skill competencies is 42 from 140 vocational schools, both public and private. The condition of vocational school buildings as of 2021 is 30.66% in good condition, 67.59% in lightly/moderately damaged condition, and 1.75% in heavily damaged condition. Central Kalimantan Province is the province that is the largest area in Indonesia, with an area of 153,564.50 km$^2$. Some places have varying areas and levels of population density [5]. From 2017 to 2022, the population in Central Kalimantan Province generally increased from 2,605,274 to 2,741,100 people, with a growth rate of 1.84% in the 2010–2020 period and 1.51% in the 2020–2022 period. The number of residents aged 10–14 years or prospective users of vocational school services tends to decrease from 2017 to 2022 by 231,695 to 229,300 people, and the population aged 15–18 years, who are the age of vocational school students, tends to increase from 2017 to 2022 by 224,219 to 236,300 people. In the 2017–2022 period, Central Kalimantan Province's gross regional domestic product (GRDP) increased from 125,817.10 billion rupiahs to 199,947.90 billion rupiahs. In the 2017–2022 period, the number of workers in Central Kalimantan Province increased from 1,222,707 to 1,344,475 people. On the other hand, the number of unemployed also increased from 53,962 to 59,829 people. The number of poor people in Central Kalimantan Province increased from 2017 to 2022 from 139,160 to 144,520 people, with the poverty line value increasing from IDR401,537 to 580,113 [6–11]. The following is Figure 1, regarding graphs related to population, GRDP, employment, and poverty.

There have been good practices in the past that have succeeded in increasing SPR indirectly. In the period from 1973 to 1978, President Soeharto issued a presidential instruction regarding the Elementary School Development Assistance Program [12–15], also known by the name SD Inpres. The regulation contains instructions for establishing many new school units/unit sekolah baru (USB) in the form of new classrooms (CR) along with supporting facilities and infrastructures, such as teachers' rooms, toilets, school furniture, reading books, clean water sources, and official housing for heads of state, schools and teachers for schools in remote areas, and the appointment of new teachers. After the policy was implemented, one of the Nobel laureates proved that the presidential instruction was a successful policy implementation from the government of Indonesia in improving education in Indonesia and increasing the income of the Indonesian population [16,17]. Regarding the establishment of SMKs, through applicable regulations, it is necessary to add at least one infrastructure to become a place of practice, namely the vocational practice room (VPR)

and vocational practice tools (VPTs) [18,19]. Several laboratories need to be provided, but if all are provided, it will require large funds. These good practices can be used as a reference for the Central Kalimantan provincial government in making policies to increase SPR. Education systems simultaneously perform three functions: they provide skills to the labor market, they are a source of public sector employment, and they distribute educational opportunities, as this study describes in the analysis projection of the last function [20].

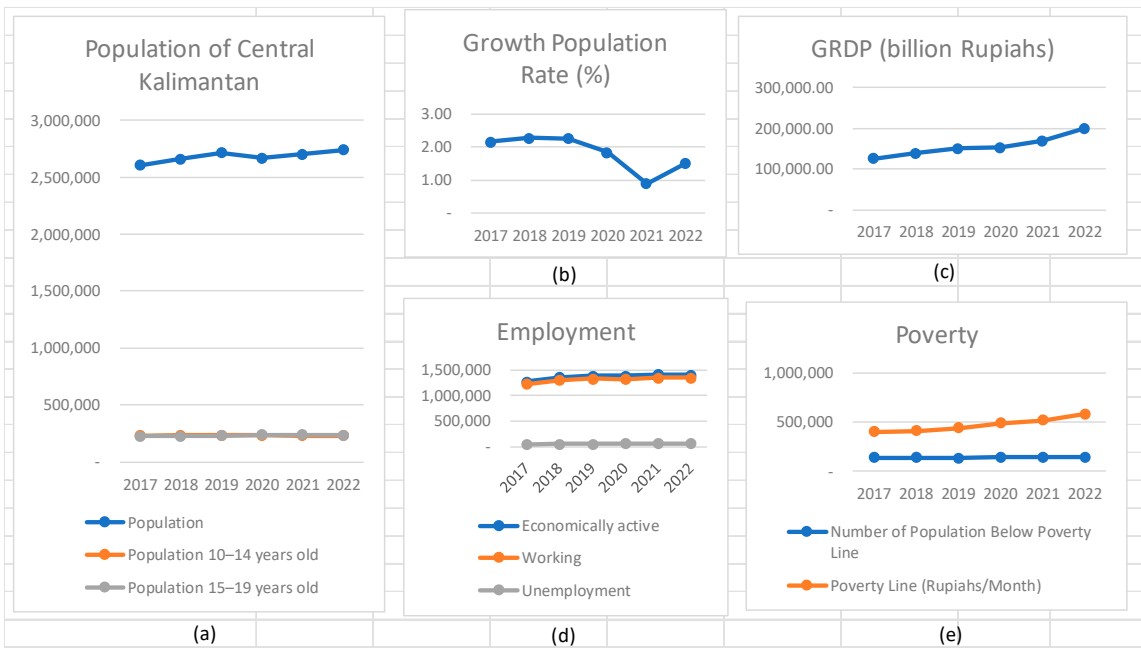

**Figure 1.** (**a**) Population of Central Kalimantan from 2017 to 2022; (**b**) population rate growth in Central Kalimantan; (**c**) gross regional domestic product of Central Kalimantan from 2017 to 2022; (**d**) number of employed people in Central Kalimantan from 2017 to 2022; (**e**) number of population below poverty and the poverty line in Central Kalimantan from 2017 to 2022.

When it comes to SMK facilities and infrastructures, the central government has set standards for SMK facilities and infrastructures through the Minister of Education and Culture Regulation/Peraturan Menteri Pendidikan dan Kebudayaan (Permendikbud) Number 34 of 2018 concerning national standards for vocational high school education. The regulation states that to achieve the minimum criteria for SMK facilities and infrastructures, a school should have facilities and infrastructures in the form of land, buildings, general learning rooms, practice rooms/general laboratories, practice rooms/expertise laboratories, leadership and administration rooms, and respective supporting rooms, each with a specific minimal size, design, and layout [19]. However, the researcher did not describe the minimum size, design, and layout because, in general, all facilities and infrastructures will be built according to the size and design of the previous building, and the layout will follow the existing site plan. Not all facilities and infrastructures by Permendikbud No. 34 of 2018 will be analyzed. Only priority facilities and infrastructures will be analyzed based on SD Inpres policies and regulations related to the establishment of SMKs. The facilities and infrastructures are classrooms (CR), teachers' work rooms, toilets, reading books, vocational practice rooms (VPRs), and vocational practice tools (VPTs). We ignore other facilities and infrastructures, such as clean water sources and official housing for SMK heads and teachers because these are facilities and infrastructures to support the establishment of new school units (USB). Likewise, with the availability of books, it is not included as a variable that must be calculated because, currently, the procurement of books can be procured using school operational assistance/bantuan operasional sekolah (BOS) funds. The author will only calculate the lack of facilities and infrastructures in existing schools. There may still be areas that do not have a secondary education unit, but it is not

certain that their residents need SMKs because one of the conditions for establishing an SMK is the existence of industry and the world of work/industri dan dunia kerja (Iduka) around the SMK area. Apart from that, this article will only describe its existence because it refers to the DPA; each facility and infrastructure, in particular, already has its unit price, and the unit price of the existing DPA is used as the basis for the analysis.

The literature that is presented is the literature related to the provision of funds and budgets by the Central Kalimantan provincial government; related to the impact of people continuing education; the need to prioritize certain facilities and infrastructures, which have a direct impact on increasing school enrollment rates; and the need for vocational education for students. The literature makes it necessary to analyze the fulfillment of vocational school facilities and infrastructures. With the same keywords, there are other studies that discuss vocational education, such as the equal distribution of opportunities to attend vocational schools [21]; dual qualification academic–vocational education in high school [22]; local revenue and parents' money, which contribute to improve the quality of education [23]; the relationship between capitalism and education [24]; or the impact of the brain drain and the benefits of circular migration that have an effect on education policy and the growing involvement of non-governmental organizations and initiatives in introducing a more practice-based orientation [25]. However, all of these studies are not sufficiently related to the objectives of this study.

Studies discussing education policy provide research results that can be used as policies and offer solutions to overcome problems that prevent children from continuing to secondary education or an SMK. Several research results related to the need for policies to increase the education budget include the following: countries that guarantee secondary education in their country's constitution have a higher NPR for secondary education (8.4 percent) than other countries that do not guarantee secondary education in their country's constitution [26]. Furthermore, other studies have concluded that there is a close relationship between local governments, where providing higher funding for education spending can prevent children from dropping out of school [27]. Regarding subsidies, an increase in the subsidy budget by 20% increases 0.21 years of schooling [28]. Regarding cash transfers, the existence of such programs has increased the continuing education rate by 60% for conditional assistance and increased the continuing education rate by 18–25% for unconditional assistance [29]. In Kenya alone, efforts to increase the number of secondary schools increase junior high school students going on to secondary school. The expansion of secondary education policies increased women's educational attainment by 0.75% and secondary school completion by 6–10% [30]. From some of these studies, it can be understood that education requires a sizable budget.

The results of research that show the impact that someone has if they continue their education include Gambia, Kenya, and other sub-Saharan countries, where free schooling ensures that women continue to pursue secondary education [31,32]. Low-income families perceive school as a place to guarantee their children's future by getting a job or going to a university [33]. The broad impact on demographic behavior of educational policies occurs by decreasing the probability of a first marriage at the age of 16 and 18 by around 40%–20%, respectively; the probability of giving birth to a first child at the age of 19 and 20 years by 30%; and the probability of first engaging in sexual intercourse at 20 years old by as much as 10%. In addition, the policy altered short-term labor market outcomes, increasing the likelihood of skilled jobs by 28% and decreasing the likelihood of agricultural jobs by nearly 80% [30]. If governments can provide an inclusive education policy, then people will have low barriers and can continue their education [34]. In some countries, such as Brazil, India, South Africa, and the United States, an inclusive education policy can promote access to higher education for diverse students [35].

Research from Duflo in 2001, which emphasized the need for equity in education by increasing capacity, has been proven to be recognized by the world community, where in 2019, thanks to the results of his research, Duflo was awarded the Nobel Prize. The research conducted, namely linking the relationship between education and poverty reduction,

stated that the more schools, the higher the wages of the community. New schools increase capacity in densely populated settlements and reduce the distance to schools in sparsely populated settlements. The positive effect is that Indonesia's gross domestic product (GDP) growth has increased rapidly due to the effect of the presidential instruction for schools. In addition, public schools provide free educational services to poor students [16]. Therefore, to prevent poor students from continuing to secondary school, the capacity of vocational high schools must be sufficient for them. In line with this, the president of Indonesia has issued Presidential Instruction Number 9 of 2016 concerning the revitalization of vocational high schools. One of the instructions to the governor of Central Kalimantan is to provide adequate vocational school facilities and infrastructures and make it easier for people to obtain vocational school education services [36]. This presidential instruction could be momentous for Indonesia, including Central Kalimantan Province, to reform vocational schools under its authority. Referring to vocational education reforms in Norway and Sweden in the 1990s [37], Indonesia and Central Kalimantan Province can experience the same success.

The results of research related to the need for vocational education for graduates of junior high school students include the following: vocational education is the best way to achieve academic success for residents aged 16 years; for those who have low academic abilities, the best way is to continue to secondary education; and for men, they will be more successful if they continue to study vocational training [38]. Vocational education is even one of the fastest ways for rural communities to get jobs. In Australia, inefficiencies at the TAFE Institute occur when providing services to students with a low economy and who are in remote (rural) areas, or it can be said that the cost per student is greater than the cost to students in urban areas and/or with economic backgrounds [39].

From the literature, it can be seen that the greater the budget for education, including vocational education, the higher the SPR for secondary education. From 2017 to 2022, there is still a gap between junior high school graduates and secondary education graduates, both general and vocational. Therefore, an in-depth analysis is needed to be able to find out how long the Central Kalimantan provincial government can fulfill vocational education infrastructures to increase SPR and realize equitable vocational education. Related to this problem, the budget allocation for SMK facilities and infrastructures must include priority facilities and infrastructures first because if it refers to the standard of SMK facilities and infrastructures in Permendikbud No. 34 of 2018, quite a lot of facilities and infrastructures must be met, and there are still several SMKs that experience a shortage of priority facilities and infrastructures. With the fulfillment of priority vocational facilities and infrastructures, it will bring the school-aged population closer to school in rural areas and increase the capacity of vocational schools in urban areas.

An SMK is a great way to improve the economy of students and their families, especially for men. The resulting externalities are also positive, including preventing children from dropping out of school and the consequences thereof, increasing people's income, thereby increasing the region's (GDP); increasing educational attainment and school completion; decreasing the rate of early marriage, giving birth at an early age, and sexual relations at an early age; and improving the skilled labor market.

By looking at the 2018 SMP SPR in Central Kalimantan Province, it is still not the same as the SPR for secondary education, so it can be assumed that there is still a shortage of facilities and infrastructures related to access to secondary education, including SMKs. Therefore, with the existing budget trends from 2017 to 2022, with information on the availability of data on facilities and infrastructures originating from Dapodik application and the number of facilities and infrastructures according to the standards stipulated in Permendikbud No. 34 of 2018, it will require a lot of budgets where not all of these facilities and infrastructures are related to increasing SPR.

The research question of this article is as follows: can the provincial government of Central Kalimantan succeed in providing SMK priority facilities and infrastructures in Central Kalimantan Province, and in what year did the provincial government achieve stan-

dardized facilities and infrastructures? The research aims to find out whether the provincial government of Central Kalimantan can meet SMK priority facilities and infrastructures and to know the year of achievement so that the provincial government can design better facility and infrastructure policies.

The calculated facilities and infrastructures are from the SD Inpres policy that was studied by Duflo who received the Nobel Prize for his research. The facilities and infrastructures are classrooms (CR), teacher work rooms (TWR), and toilets. Other facilities and infrastructures, such as reading materials, are not included in the calculation because reading materials can be financed from the BOS fund; at present, the education office considers that the responsibility for providing BOS funds is handed over to schools to divide the responsibility for provision so that the focus of the education office is to provide facilities and infrastructures, which cannot be financed from BOS funds alone. Apart from that, there is the provision of official housing, but currently, there is insufficient information regarding the amount of official housing and its condition, the amount of the budget needed to procure official housing, and which schools are entitled to provide official housing.

Based on the rules for the establishment of an SMK, it must have practice space for practice as well as practical tools because the competence of graduates expected from an SMK is to be skilled at working on a piece of work. Therefore, the facilities and infrastructures that are counted are vocational practice rooms (VPRs) and vocational practice tools (VPTs). However, not all schools record vocational practice tools or provide information for where the vocational practice tools are located. In addition, the Regulation of the Minister of Education and Culture does not explicitly stipulate standard names, types, and numbers of vocational practice tools for each spectrum, so researchers find it difficult to calculate the need for vocational practice tools. Furthermore, the researchers also found it difficult to calculate the need for budget proposals, considering there was no unit price information for each practical tool for students in each department and each region, which had various names, types, and amounts. Therefore, the variable of vocational practice tools will still be included in the model as a part of priority facility schools.

This study only describes the relationship between priority facilities and infrastructures that are directly related to increasing SPR. There are several conditions that influence the increase in SPR, such as restrictions due to COVID-19 or the value of agricultural products, but this study will not describe these factors. The research will only describe the relationship between the budget for facilities and infrastructures in the Central Kalimantan provincial government and the provision of priority facilities and infrastructures, which is a physical feature and one of the factors that influences the increase in SPR.

## 2. Materials and Methods

### 2.1. Research Design and Strategy

The research method in this article used a quantitative method with a system dynamic model approach. Quantitative research is a set of interrelated constructs (or variables) formed into propositions or hypotheses, which determine the relationships between variables that are useful in helping to explain phenomena [40]. To create a model, the author used the vensim PLE64 application version 9.2.0. Vensim is a modeling tool for creating and simulating dynamic models. Vensim modeling is often used in business, science, and education. Vensim can sketch causal loop models, but to simulate them, it is necessary to create equations [41]. The business process of fulfilling SMK facilities and infrastructures and the policies that affect the fulfillment of facilities and infrastructures are presented in a system dynamic model. The system dynamic model itself is described in the image of a feedback loop where decision makers/policy makers compare information about real-world conditions with various goals, feel the difference between the desired state and the actual state, and make decisions that are believed to cause a real condition to be in the desired state [42].

Some principles for modeling systems dynamic according to Sterman [42]:

1.  The input to all decision rules in the model must be limited to information that is available to the actual decision maker.

    a.  The future is unknown to anyone. All hopes and beliefs about the future are based on historical information. Therefore, expectations and beliefs may be wrong;

    b.  Actual and perceived conditions differ due to measurement and reporting delays, and beliefs are not updated immediately after receiving new information. Perceptions often differ from actual situations;

    c.  The outcome of untried contingencies is unknown. Expectations about "what if" situations that have never been experienced are based on known situations and may be wrong.

2.  Model decision rules must conform to managerial practices.

    a.  All variables and relationships must have real-world equivalents and meanings;

    b.  The units of measurement in all equations must be balanced without using arbitrary scaling factors;

    c.  Decision making should not be assumed to conform to a prior theory but should be directly investigated.

3.  Desired conditions and actual conditions must be distinguished. The physical constraints for the realization of the desired result must be represented.

    a.  The desired state and the actual state must be distinguished;

    b.  Desired and actual levels of change must be distinguished.

4.  Decision rules must be robust under extreme conditions.

5.  Equilibrium must not be assumed. Equilibrium and stability may (or may not) arise from the interaction of system elements.

From the principle of making the model, it will then be made in the form of a stock-flow diagram, where all values are entered in each variable based on the data that has been collected, then simulations are carried out for several policy scenarios, the simulation results are presented, and the results of several simulations are analyzed.

*2.2. Data Collection*

The data used came from school profile data, which can be downloaded from the Dapodik application for each SMK. Dapodik is the information system that collects whole data about schools, such as infrastructures, facilities, students, teachers, and classes. Dapodik operator in each school inputs the data, and everyone who has username can access it. The data regard what and how many facilities and infrastructures each school has. In addition, there is information on study groups, number of students, number of teachers, and number of experts to determine the ideal number of classrooms, toilets, teachers' work rooms, and vocational practice rooms (VPRs). In addition, the data source needed was the DPA of the education office in PSMK (Pembinaan Sekolah Menengah Kejuruan/Vocational High School Management) Division, which contains the budget allocation for each facility and infrastructure. Here is the PSMK DPA between 2017 and 2022 in Table 1 and Figure 2.

The school profile data reliability test was carried out by reconfirming several facilities and infrastructures owned by the school to the school principal or application operator, whose number and extent were anomalous. For budget data, the researcher tested it by comparing the DPA with the value input of facilities and infrastructures carried out by the management staff of SMK facilities and infrastructures.

The number of SMKs to be counted was 138 out of a total of 140 public and private SMKs, which was based on the number of SMKs that had provided data on their facilities and infrastructures. The full responsibility for fulfilling facilities and infrastructures by the provincial government of Central Kalimantan is only on the public SMKs. However, private SMKs that experience a shortage of facilities and infrastructures need help because the foundation that manages private SMKs aims to help the government so that every school-aged population can attend school.

**Table 1.** Amount in PSMK Division Budget 2017–2022.

| | | Kind of Budget | |
|---|---|---|---|
| | | **Approved Budget PSMK** | **Approved Budget PSMK Non-Facilities and Infrastructures** |
| Amount (Rupiahs per Year) | 2017 | 81,680,000,000 | 41,987,730,010 |
| | 2018 | 67,603,400,663 | 67,112,390,900 |
| | 2019 | 131,661,314,500 | 2,298,588,500 |
| | 2020 | 138,003,990,220 | 38,174,268,500 |
| | 2021 | 135,110,545,500 | 69,679,992,150 |
| | 2022 | 130,168,898,165 | 42,050,000,000 |
| Growth per Year | (2022–2017)/5 | 11.87%/year | 13.14%/year |

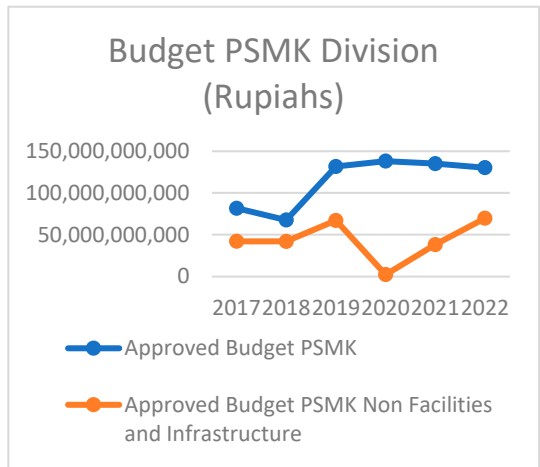

**Figure 2.** Budget PSMK Division.

Based on the DPA, if the SMK facility and infrastructure budget data have a trend that tends to increase and if the amount in the priority facility and infrastructure budget is higher than the budget for non-priority facilities and infrastructures, then the priority education facilities and infrastructures should be achieved. Apart from that, with the dynamic system method, we carried out various existing scenarios to determine vocational school facility and infrastructure policies that were more appropriate and more implementable to the conditions of SMKs in Central Kalimantan.

## 3. Results

### 3.1. Facility and Infrastructure Business Process

Facilities and infrastructures (SP) are countable items that can increase and decrease. The addition of one item of facilities and infrastructures comes from the procurement of goods and services, which are carried out either through self-management or at auctions to providers (SPb). Additions can come from grants from both parents and other local governments, but in the last five years, there have been no additions from these two things. Therefore, the factor of adding facilities and infrastructures comes from government spending.

Each item may be reduced in number. For facilities and infrastructures, the number may decrease due to heavy damage (SPrb). The factor affecting damaged buildings is the useful life (MM). The useful life of the building is 50 years according to Government Regulation No. 16 of 2021, concerning government buildings. This means that, in one year, it is estimated that 1/50 of the damaged infrastructure or 0.02 times the existing infrastructure is damaged. The current condition of the infrastructure is not as a new item that will be damaged in 50 years, but some have been established for several years,

so the number of damaged infrastructures is, of course, not uniform in the number per year. In addition, the infrastructure can last for more than 50 years if properly maintained, or they can be damaged more quickly due to natural disasters (high winds, fires, etc.). However, information for these two matters is not included in the school profile data, so the assumption is that the useful life of the infrastructure used is 50 years.

The relationship between facilities and infrastructures (SP), addition ($SP_b$), and subtraction ($SP_{rb}$) is described in the following formula:

$$SP = SP_b - SP_{rb} \tag{1}$$

Furthermore, the needs (B) of each facility and infrastructure are calculated based on the difference between the ideal number of facilities and infrastructures ($SP_i$) and the current number (SP) plus the number of damaged facilities and infrastructures ($SP_r$) in the same year. Therefore, with this explanation, the following formula is obtained:

$$B = SP_i - SP + SP_r \tag{2}$$

The amount needed for the budget for facilities and infrastructures (BA) is determined depending on the unit price (HS) of each facility and infrastructure each year. Based on this explanation, the following formula is obtained:

$$BA = B \times HS \tag{3}$$

The need for each type of priority facility and infrastructure, such as classrooms ($BA_{rk}$), vocational practice rooms ($BA_{vpr}$), toilets ($BA_j$), teacher rooms ($BA_{rg}$), and vocational practice tools ($BA_{aps}$), will be added together to form the proposed priority needs for facilities and infrastructures (UBA). Based on this explanation, it is formulated as follows:

$$UBA = BA_{cr} + BA_{vpr} + BA_j + BA_{twr} + BA_{vpt} \tag{4}$$

In the budget for the SMK coaching program ($UBA_k$), there are several budgets outside of priority facilities and infrastructures, but the researchers divided them into two, namely the budget for non-priority facilities and infrastructures ($BA_{spt}$) and the budget that is not for facilities and infrastructures ($BA_{ssp}$). This is due to the policy of dividing budget allocations for all facilities and infrastructures. Based on this explanation, it is formulated as follows:

$$UBA_k = UBA + BA_{spt} + BA_{ssp} \tag{5}$$

After making a budget proposal, the provincial government of Central Kalimantan, together with the DPRD (Dewan Perwakilan Rakyat Daerah/House of Local Representatives), determines the temporary budget ceiling in advance at the PPAS (Proritas Plafon Anggaran Sementara/Provisional Budget Ceiling Priority) meeting, whose activities often coincide with KUA (Kebijakan Umum Anggaran/General Budget Policy) meetings. This ceiling is the reference, which is the maximum amount of the budget given by each work unit, to then select schools that prioritize getting facilities and infrastructures. It is appropriate that the distribution of the budget is based on needs, but what happened is that the amount of PPAS for the SMK development program (budget for fulfilling facilities and infrastructures and non-facilities and infrastructures) only increased slightly from the previous year's budget. Even for other programs, it still happens when the needs are met. Therefore, the budget for facilities and infrastructures that has been approved is the SMK development budget, which has increased slightly from the previous year and needs to be reduced in the budget for aspects other than facilities and infrastructures. However, if the budget requirements for priority facilities and infrastructures are smaller than the previous year's budget, then the budget for priority facilities and infrastructures is as large as the budget requirements, and the remainder can be used to meet other facilities and infrastructures or activities outside of meeting the needs of facilities and infrastructures.

After the PPAS is determined, the distribution of the budget ($A_{sp}$) is made for the fulfillment of classrooms, VPRs, VPTs, teachers' rooms, and toilets. The division is a separate policy, so it requires a new variable ($PA_{sp}$). This is formulated as follows:

$$SP = A_{sp} \times PA_{sp} \tag{6}$$

*3.2. Policy Model for Fulfillment of Priority Facilities and Infrastructures for Vocational Schools*

Based on this explanation, the relationships that are formed to fulfill priority facilities and infrastructures can be modeled in a causal loop diagram (CLD), as shown in Figure 3.

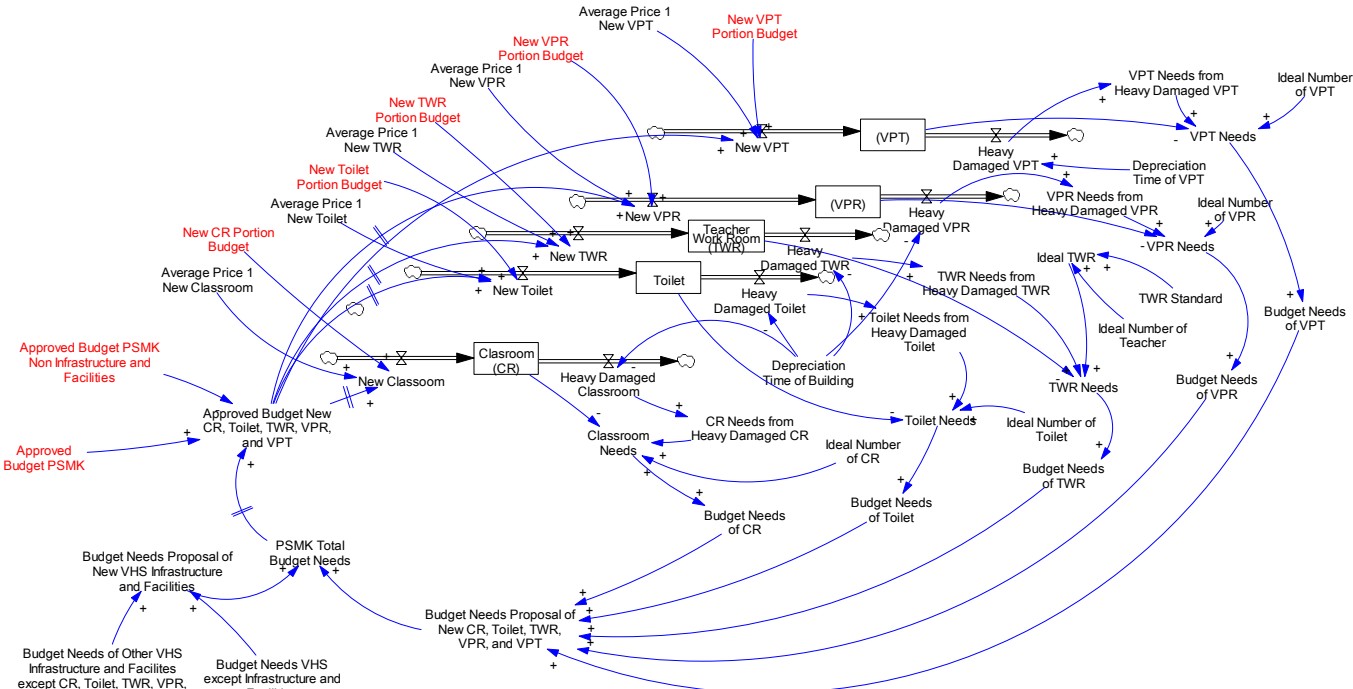

**Figure 3.** Policy model for fulfillment of priority facilities and infrastructures in vocational high schools.

In the model, it can be seen that the provincial government of Central Kalimantan is trying to fulfill the facilities and infrastructures according to their respective ideal numbers. The gap between the ideal amount and the current amount creates fulfillment needs, and these needs are converted into budget requirements. However, the need for the budget is not realized by dividing the portion of the ideal budget. Annual budget approval is only determined by policies that apply the same from year to year between the provincial government and the provincial DPRD (Local Legislative Assembly), namely based on the absorption of the budget in the previous year, and the amount can be increased slightly. Even then, the budget is a combination of all vocational training programs. Therefore, the size of the SMK coaching program budget for the fulfillment of facilities and infrastructures also depends on budget policies for activities outside of priority facilities and infrastructures, both for non-priority facilities and infrastructures and for activities outside of facilities and infrastructures. The proportion of the budget for fulfilling priority facilities and infrastructures is also determined by the policy of dividing the budget between priority facilities and infrastructures. This is because the condition of each SMK is different from one another. Ideally, SMKs that have a gap between the current number and the ideal number will be given a larger budget. However, the influence of political, social, cultural, geographical conditions, and population growth is quite influential in determining the proportion of distribution, including lobbying between parties to try to get more budgets. It should be noted that the fulfillment of school facilities and infrastructures is also an

activity of procuring goods and services, which is the cause of high cases of corruption in Indonesia.

*3.3. Scenario Fulfillment of Priority Facilities and Infrastructures for Vocational Schools*

3.3.1. Total Budget for Priority Facilities and Infrastructures Remains Allocated According to Current Trends

To obtain information on the achievements of the Central Kalimantan provincial government in fulfilling priority facilities and infrastructures, a simulation was carried out based on the education profile data of each SMK, DPA data from the education office from 2017 to 2022, and budget trends from 2017 to 2022 according to the year the Central Kalimantan provincial government started managing SMKs; then, we can carry out a simulation using the vensim application to test the implementation of the policy of fulfilling priority facilities and infrastructures based on the behavior of several variables as follows:

1. The portion of the budget for each facility and infrastructure is calculated by the percentage of budget requirements for each type of facility and infrastructure;
2. The growth of the PSMK Division budget per year is 11.87% per year;
3. PSMK Division budget growth, excluding priority facilities and infrastructures, is 13.14% per year;
4. The growth in unit prices for classrooms, VPRs, teacher rooms, and toilets is 8.5% per year;
5. There is VPT information in the school profile data, but there are no rules for the number, type, and name of the ideal VPT for each skill and for each school and the unit price per VPT. Therefore, the number of VPTs and the ideal number of VPTs are expressed as 0 units. This is because the simulation is only carried out for variables that have complete data. For this reason, no VPTs are badly damaged, and there are no figures for VPT needs, new VPT needs, and VPT budgeting needs.

Based on the simulation results, as shown in Figure 4, it is known that the gap between the proposal and the approved budget for the budget for priority facilities and infrastructures will be the same starting in 2026. This indicates that the provincial government of Central Kalimantan only needs to maintain consistency in the policy of fulfilling the budget until 2026. After, when the value between the proposal and the approved budget is the same, the remaining budget can be diverted to fulfill other facilities and infrastructures.

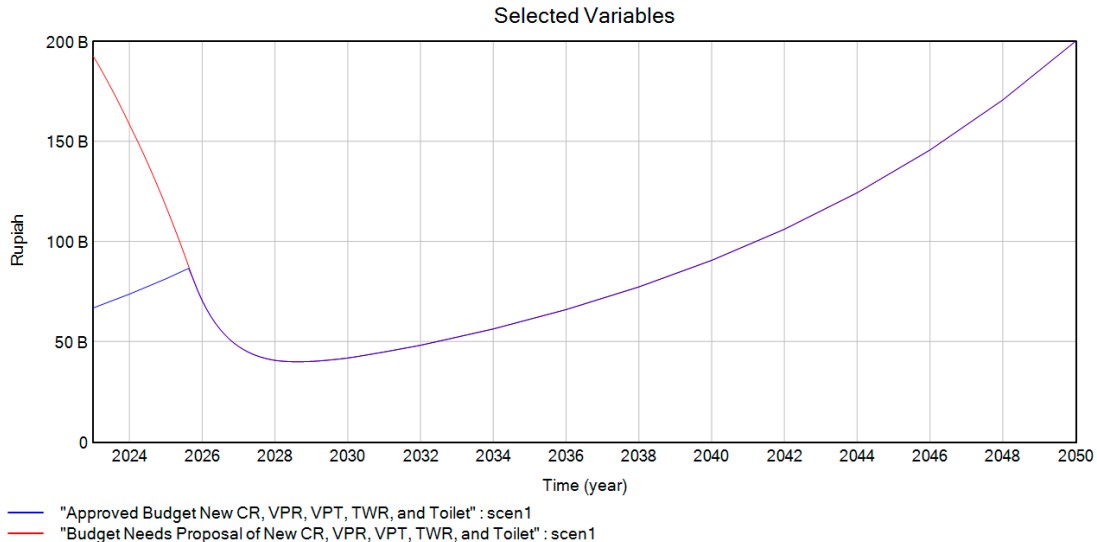

**Figure 4.** Budget simulation results for scenario 1.

On the other hand, based on the simulation results in Figure 5, it is known that the number of each infrastructure, including classrooms, VPRs, teacher rooms, and toilets, will be the same as the ideal number in 2028. This difference occurs because, in the year the budget for fulfilling facilities and infrastructures was realized, at the same time, there

is the infrastructure that was badly damaged, as shown in Figure 6. However, this will not last long because, in 2028, the number of the infrastructures and facilities required will be the same as the number needed for damaged facilities and infrastructures. Of course, as previously reported, the number of VPTs was declared 0 due to the absence of comprehensive data.

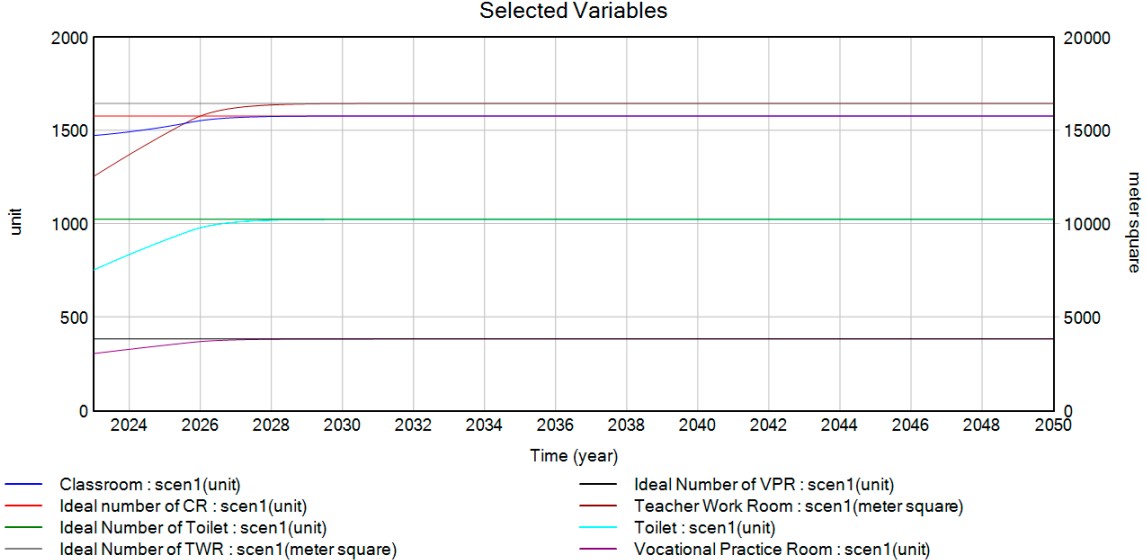

**Figure 5.** The simulation results for meeting the ideal number of facilities and infrastructures.

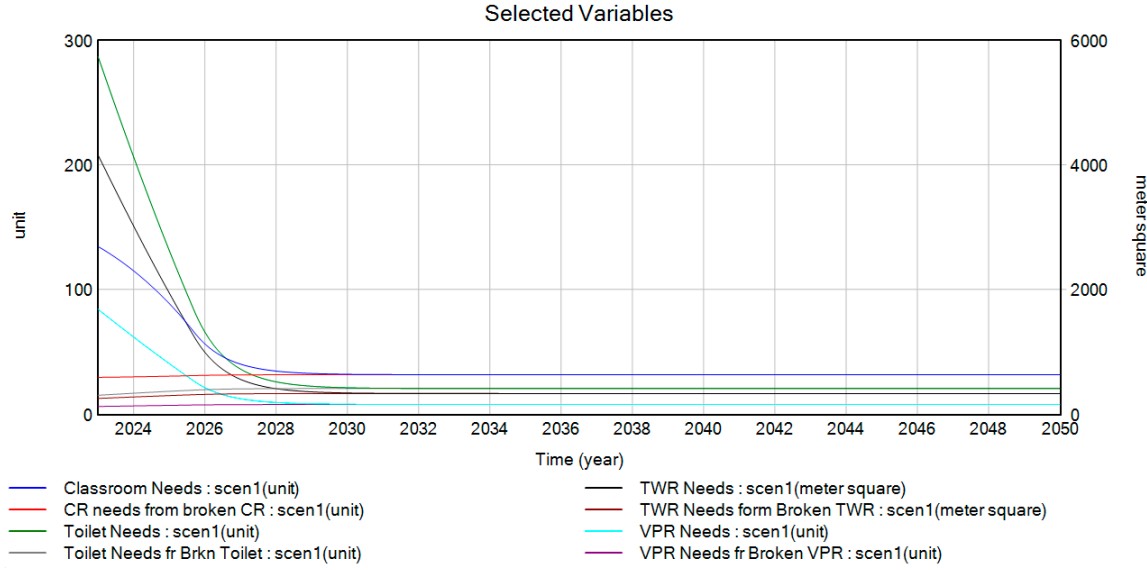

**Figure 6.** Simulation results for facility and infrastructure needs.

### 3.3.2. Scenario Total Budget for Facilities and Infrastructures Allocated for Priority Facilities and Infrastructures

Based on the behavior defined in point 3.3.1, in this scenario, the budget for facilities and infrastructures obtained is larger because the budget for facilities and infrastructures for maintenance and non-priority facilities and infrastructures is diverted to priority facilities and infrastructures. In addition, the budget growth outside of facilities and infrastructures amounted to $-1.49\%$, so budget growth only occurred in the priority facility and infrastructure budget.

Based on the simulation results, as shown in Figure 7, it is known that the gap between the proposal and the approved budget for priority facility and infrastructure budgets

will be the same starting in 2024 (or two years faster). On the other hand, based on the simulation results in Figure 8, it is known that the number of classrooms, VPRs, teacher rooms, and toilets will be the same as the ideal number in 2026, where the difference in achievement with scenario 1 is two years faster. This can be an attractive option for the Central Kalimantan provincial government because the governor of Central Kalimantan is currently entering his second term of government, which will end in 2026. If the current governor wants to show success, which is that, during his reign, there were no prospective SMK students who were rejected due to a lack of capacity, this strategy needs to be considered.

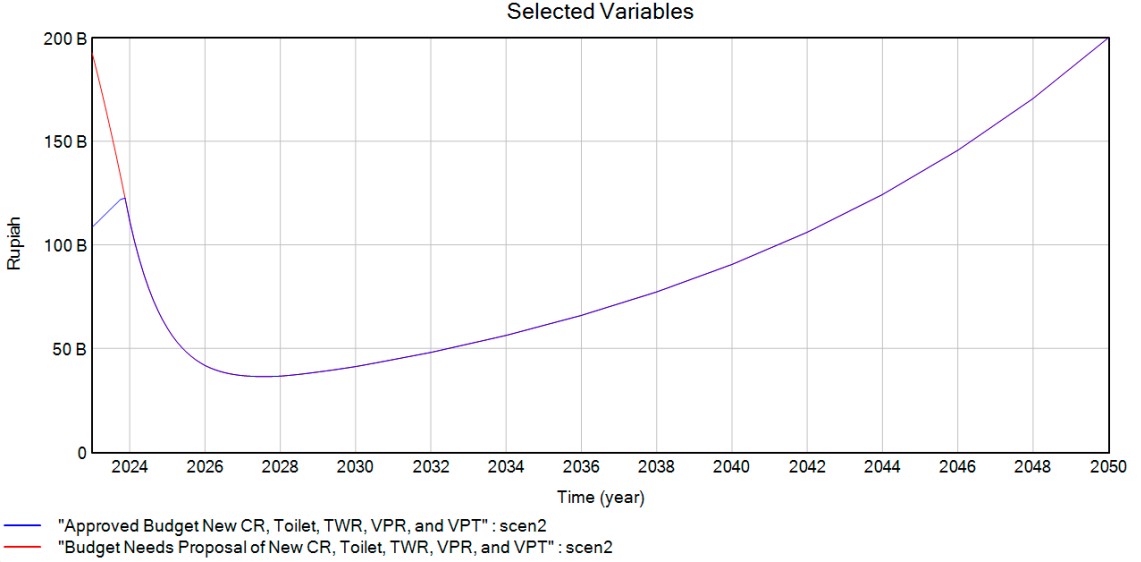

**Figure 7.** Scenario 2 budget simulation results.

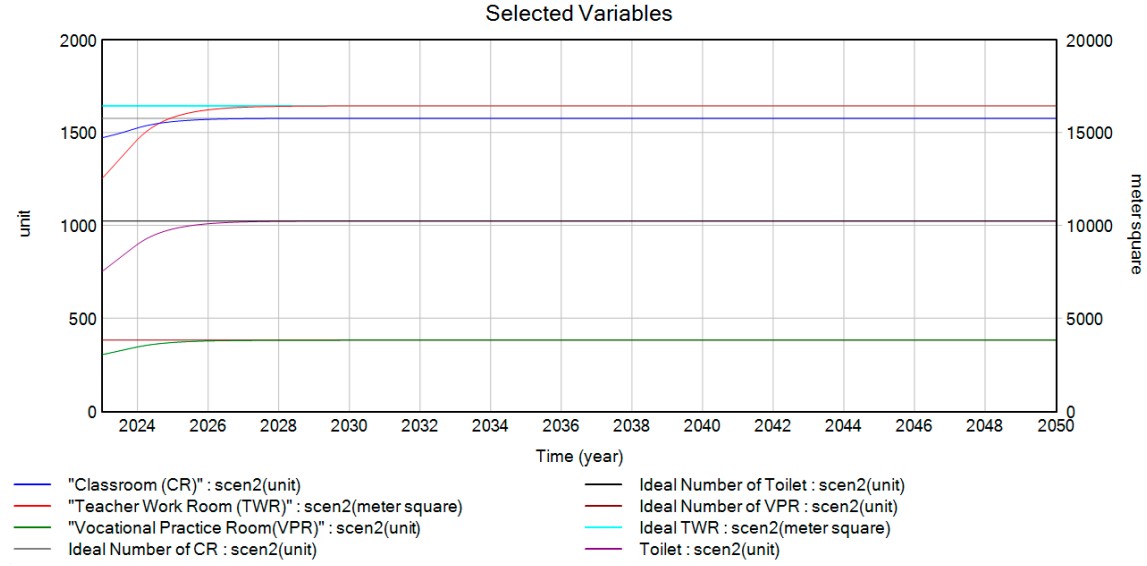

**Figure 8.** Simulation results of fulfilling scenario 2 facilities and infrastructures.

It is important to note that facilities and infrastructures may be seriously damaged, but management will not allow them to be damaged without maintenance. When a facility and infrastructure are moderately damaged, it is appropriate for the infrastructure to be maintained. Therefore, the need for heavily damaged infrastructures is the need for facilities and infrastructures that have suffered heavy damage and will be maintained.

### 3.3.3. Scenario of Increasing Priority Infrastructure and Facility Budget Exceeding Current Trends

Based on the behavior that has been defined in point 3.3.1, in this scenario, the trend/acceleration of the increase in the budget for facilities and infrastructures obtained is two times larger than the previous year, namely 23.75% from the previous year. Based on the simulation results, as shown in Figure 9, it is known that the gap between the proposal and the approved budget for priority facilities and infrastructures will be the same starting in 2024 or two years faster than scenario 2. On the other hand, based on the simulation results in Figure 10, it is known that the number of classrooms, VPRs, teacher rooms, and toilets will be the same, and the ideal number in 2026 is also the same as scenario 2. This third scenario can also be an alternative policy that can be chosen by the governor of Central Kalimantan in informing his performance achievements. Here, we can see that if the governor wants to show success, which is that, during his reign, no prospective vocational students were rejected due to a lack of capacity, then the governor of Central Kalimantan can choose one of two scenario options, namely scenarios 2 and 3.

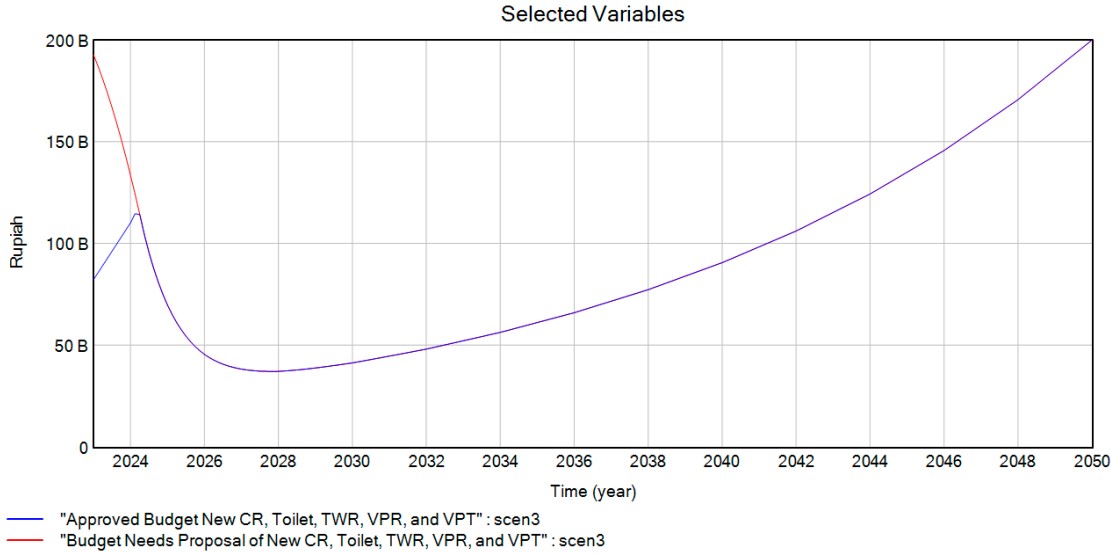

**Figure 9.** Scenario 3 budget simulation results.

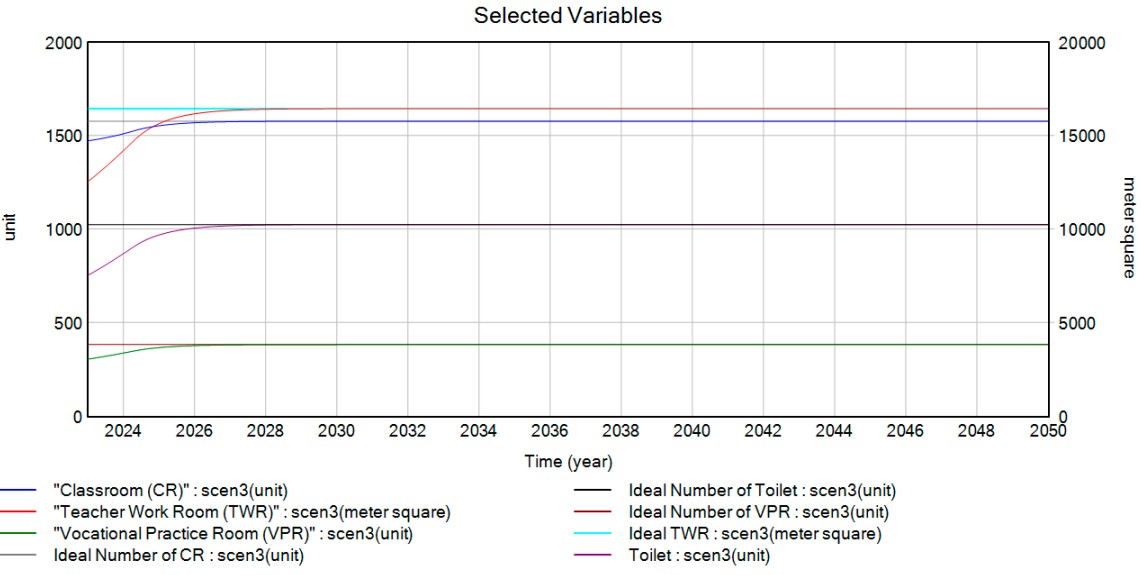

**Figure 10.** The results of scenario 3 infrastructure simulation.

### 3.3.4. Scenario Budget for Facilities and Infrastructures Used Only for Classrooms

Based on the behavior that has been defined first, in this scenario, all budgets in the current year are focused on building classrooms first, and the rest is on building other priority infrastructures. This scenario is based on the need to prioritize teaching and learning infrastructures first and then fulfill other infrastructures, such as VPRs. This scenario is suitable for conditions where local government policies focus on accepting any number of prospective students in advance so that no child is out of school.

Based on the simulation results, as shown in Figure 11, it is known that, in 2023, the approved infrastructure development budget is already greater than the budget requirements for meeting classrooms. On the other hand, based on the simulation results in Figure 12, it is known that the number of classrooms will be the same as the ideal number in 2025. This is the same as the previous scenario where, after the proposal/budgetary requirement for an infrastructure equal to the budget is approved, the number of infrastructures is equal to the ideal number in the last two years.

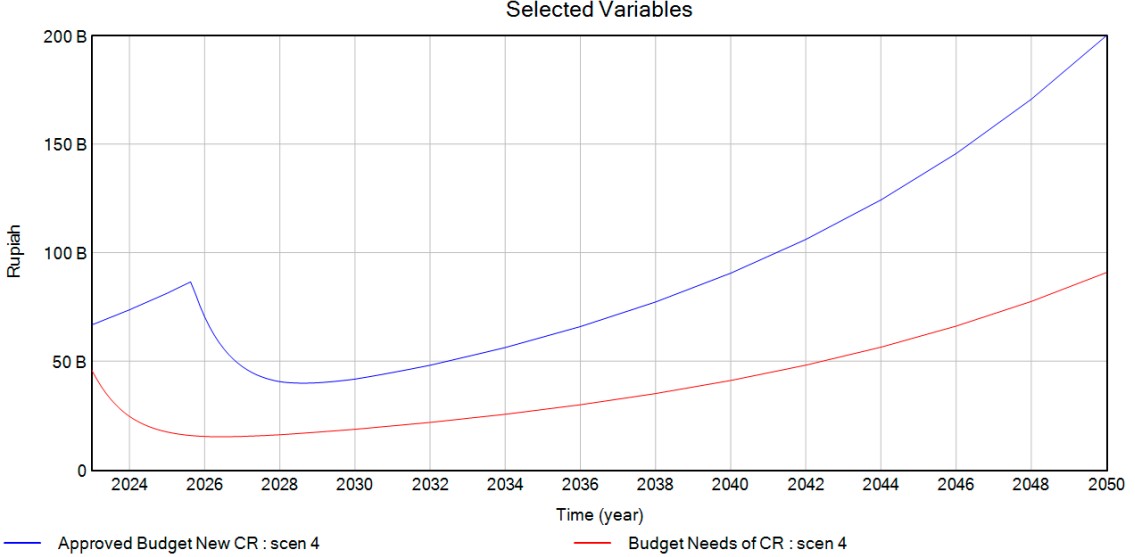

**Figure 11.** Scenario 4 budget simulation results.

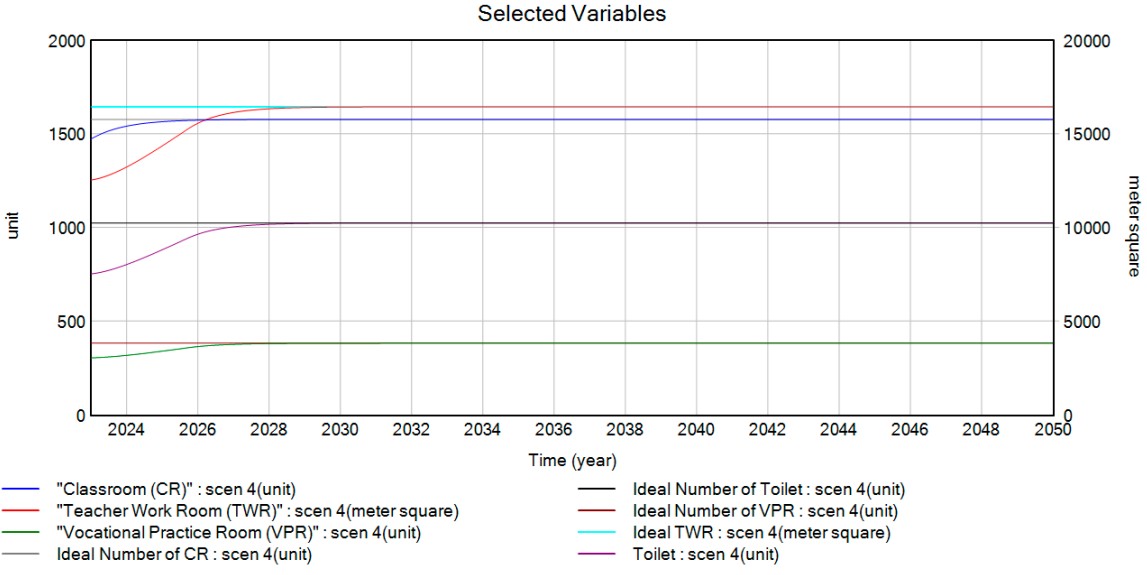

**Figure 12.** The results of scenario 4 infrastructure simulation.

For the fulfillment of other infrastructures, such as VPRs, teacher rooms, and toilets, the number will be the same as the ideal number in 2027. This third scenario can also be an alternative policy that can be chosen by the governor of Central Kalimantan in informing his performance achievements. Here, we can see that, if the governor wants to show success, which is that, during his reign, no prospective vocational students were rejected due to a lack of capacity, then the governor of Central Kalimantan can choose one of three scenario options, namely scenario 2, 3, or 4.

### 3.3.5. Scenario Budget for Facilities and Infrastructures Used Only for Vocational Practice Rooms

Based on the behavior that has been defined first, in this scenario, all budgets in the current year are focused on building VPRs first, and the rest is on building other priority infrastructures. This scenario is based on the need to prioritize practicing infrastructures first and then fulfilling other infrastructures, such as CRs. The VPR is three times more spacious than the CR, and usually, it can be separated into three to five rooms. That is enough to use one of the separated rooms for learning and teaching activities. In addition, practicing a learning activity is an advantage of SMKs over SMAs. This scenario is also suitable for conditions where local government policies focus on accepting any number of prospective students in advance so that no child is out of school, such as in scenario 4.

Based on the simulation results, as shown in Figure 13, it is known that, in 2023, the approved infrastructure development budget is already greater than the budget requirements for meeting VPRs. On the other hand, based on the simulation results in Figure 14, it is known that the number of VPRs will be the same as the ideal number in 2025. This is the same as the previous scenario where, after the proposal/budgetary requirement for an infrastructure equal to the budget is approved, the number of infrastructures equals the ideal number in the last two years. This is also the same as in scenario 4.

For the fulfillment of other infrastructures, such as CRs, teacher rooms, and toilets, the number will be the same as the ideal number in 2027. This fourth scenario can also be an alternative policy that can be chosen by the governor of Central Kalimantan in informing his performance achievements. Here, we can see that, if the governor wants to show success, which is that, during his reign, no prospective vocational students were rejected due to a lack of capacity, then the governor of Central Kalimantan can choose one of four scenario options, namely scenario 2, 3, 4, or 5.

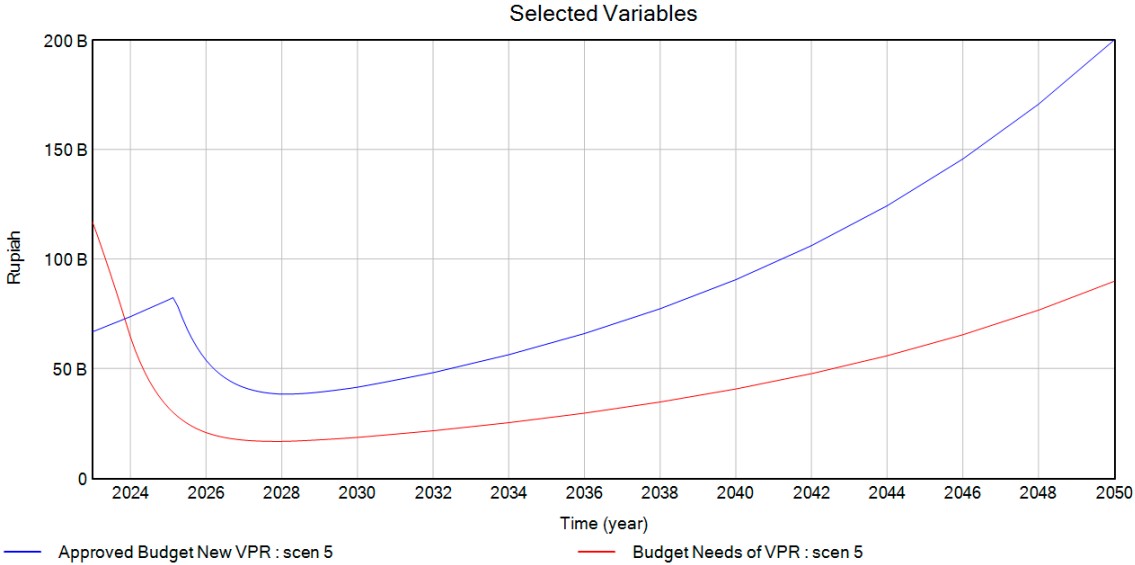

**Figure 13.** Scenario 5 budget simulation results.

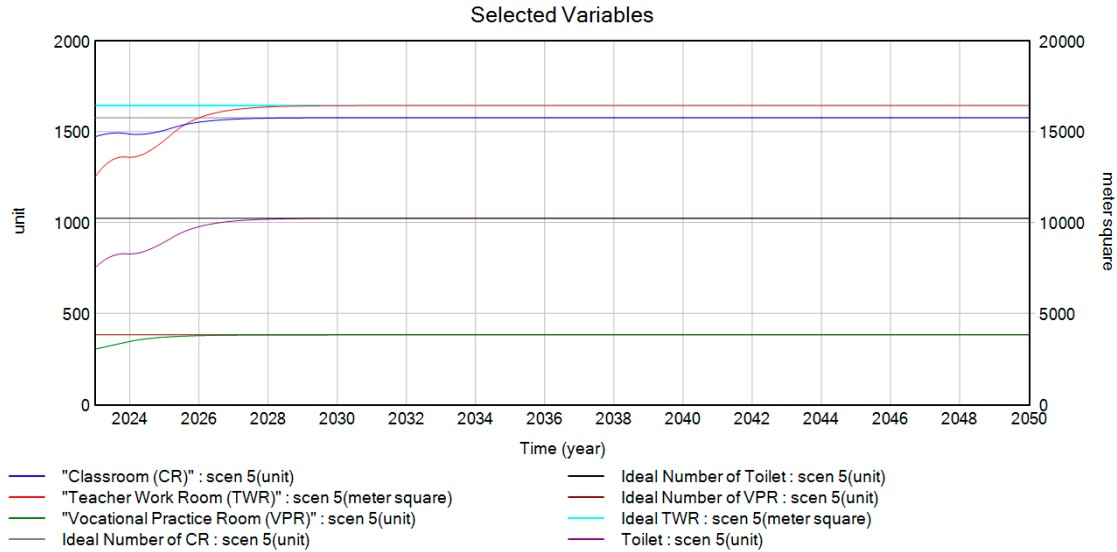

**Figure 14.** The results of scenario 5 infrastructure simulation.

## 4. Discussion

Based on the simulation results, there are several results that can be an attractive choice for the Central Kalimantan provincial government because the governor of Central Kalimantan is currently entering his second term of office, which will end in 2026. Some prospective vocational students are rejected due to a lack of capacity, so scenarios 2, 3, 4, and 5 require consideration. Previous research conducted by Duflo presented the positive impact of elementary school policies that have few types of facilities and infrastructures. In addition, other studies discuss the positive impact of education. Therefore, school facility and infrastructure policies need to be applied to SMKs as well. Moreover, an SMK is one of the fastest routes to get a job because its graduates are considered skilled according to their competence. Other research has linked a positive relationship between the provision of the education budget and a high opportunity to attend school. An SMK itself has learning characteristics that emphasize a lot of practicum activities, so it requires a larger budget to provide a special room for practice along with the equipment and practice materials. Research discussing the vocational education budget recognizes that the budget for vocational education is larger than the budget for general education. Even though the budget provision for SMKs is larger than it is for public schools, it must be admitted that every student has an interest in learning. Therefore, the provincial government of Central Kalimantan must accommodate the interests and talents of every student who is only interested in SMKs.

Based on Figure 1, it can be seen that from 2017 to 2022, it can be known that the population aged 10–14 years is almost the same and tends to decrease slightly so that, in 2026 or 2028, the provision of facilities and infrastructures according to the projection results is still relevant. Other factors, such as GRDP, are expected to continue to increase considering that the trend in the number of workers is always increasing, the area of Central Kalimantan Province is quite large, and sectors related to plantations and exploration are still the mainstay [6–11]. Although further research is needed, researchers can assume that increasing the number of facilities and infrastructures will increase the number of students, which will then increase the number of vocational school graduates who will become workers; ultimately, the GRDP will increase. Apart from that, there are other factors that are anomalies. The poor population in Central Kalimantan continues to increase. However, this could come from the arrival of residents from outside Central Kalimantan Province into Central Kalimantan Province, competing with local residents for jobs.

This study only has an impact on the provision of facilities and infrastructures, which are the means by which students can attend an SMK. Several other studies can be included in this study so that the facilities and infrastructures provided can be useful, such as the

stakeholders who stressed the need for policy measures to improve social perceptions about SMKs, coordination between multiple public agencies as well as between the government and Iduka directly involved in SMKs, and the improvement of SMK curricula, associated funding schemes, and human resource capacity [43,44]; the positive contribution and cost-effective means of using teacher assistants [45]; teaching and developing the kinds of soft skills that are suitable for Iduka demands to the students [46]; and the success of e-learning policy in Saudi Arabia [47]. In addition, a graduated student can develop a plan, including thinking to be a teacher of an SMK and continuing on to study at the university [48]. Choosing an SMK, taking apprenticeship training, and completing vocational education can increase the possibilities of finding a job with good earning conditions [49]. Students should be cultivating their core professional literacy and innovativeness, which are continuously improved [50]. Interaction between Iduka and structures of SVS programs must be built and maintained, including the contribution of the programs' structure, regional featured or key industries, changes in admission policies by the local education authority [51], and the current educational policy transfer in VET, such as changes in work and competence requirements, the influence of big data on information spreading, evidence building and policy adoption, the digital exchange of information, and digital networking and knowledge building via social media [52].

However, when facilities and infrastructures are more complete, they can create a learning environment, will make the teaching and learning process more effective, and can foster interpersonal relationships between students and teachers. Facilities and infrastructures, such as learning centers and facilities related to distance learning, are also needed because quite a few students can afford to buy laptops and other electronic learning devices [53]. Likewise, at universities in Sri Lanka, there is a good positive relationship between academics and students among themselves in using technology, such as the Internet, technological devices, mobile apps, and software [54].

The results of this study provide information regarding the projected timeframe for achieving the provincial government of Central Kalimantan to fulfill SMK facilities and infrastructures. The main factor in this effort is the budget factor. A budget is also needed for needs other than facilities and infrastructures. Research is needed regarding the role of policy actors in making decisions related to budget allocations to produce better policies for fulfilling SMK facilities and infrastructures.

Priority facilities and infrastructures that have been provided by the Central Kalimantan provincial government will be useful if they are maximally utilized. Various policies can be considered by both local governments and schools. First impressions of the school climate are important, and efforts to welcome learners with positive school activities are necessary [55]. A private school is like a service that is sold to parents and learners. Therefore, it is necessary to promote private schools to shape people's opinions and convince them to choose the school. The promotion needs to be emotionally appealing and not too much about the functional benefits of schooling, such as the general teaching and learning process [56]. Students' social background and gaps in school services are factors that determine SMP graduates' choice of school. Policy makers should routinely develop and evaluate school management policies, learner development, and learner social backgrounds. Making and evaluating them should involve students because they are the recipients of educational services provided by the school [57]. At the university level, branding is needed to attract more students. School branding can attract students to enroll in a university and lecturers to teach at the university, consequently increasing productivity. School branding arises from research excellence, university ranking information, research culture on campus, and service quality [58,59]. In SMK Muhammadiyah Kandanghaur, good school management for effective resource management to achieve organizational goals is important. From this management, a cooperation strategy is created between the school and Iduka in the short, medium, and long term and conducts monitoring and improvement on these strategies. Regarding the achievement of strategies that have been made between schools and Iduka, one of them is the creation of work culture guidelines that

must be met by students, teaching materials that are aligned with Iduka's needs, teacher internships at Iduka, and guest teachers from Iduka. The thing that attracts students is the commitment to absorb graduates, but it is not easy to convince Iduka. SMK Kandanghaur also focuses on the digital marketing of services, such as providing vehicle maintenance services in the form of a teaching factory [60].

This projection has obstacles, namely the lack of sufficient information regarding VPTs, both regarding VPT naming standards, standard quantities per skill, and unit price per unit for projection purposes. VPRs without VPTs will be useless, and the building cannot be utilized to support vocational practices. This projection also does not take into account the number of prospective students who register at each SMK in an integrated manner. The population aged 10–14 years cannot be used as a reference because each population has different interests in continuing their education. Currently, the list of prospective students is stored in separate data and not in Dapodik because there are no features in Dapodik yet. The Ministry of Education and Culture as an application administrator needs to consider this feature. This relates to the increasing number of classroom needs, and the number of hours teaching teachers will also increase, so budget projections and teacher needs must also be taken into account together because the number of teachers is proportional to the number of study groups, which is the standard reference for the number of classrooms. Therefore, due to this deficiency, if the VPT data are complete and data on prospective students are already available, it is hoped that future researchers can make projections taking this information into account to obtain more accurate projections.

## 5. Conclusions

This study only focuses on projections for the fulfillment of SMK facilities and infrastructures in Central Kalimantan Province with the facility and infrastructure budget trend from 2017–2022 to assess the achievements of the Central Kalimantan provincial government regarding the fulfillment of SMK facilities and infrastructures. Based on information on school profile data and DPA data from the education office from 2017 to 2022, we can see various kinds of behavior from fulfilling facilities and infrastructures:

1. The portion of the budget for each facility and infrastructure is according to the percentage of budget requirements for each type of facility and infrastructure;
2. The growth of the SMK development budget is 11.87% per year;
3. PSMK Division budget growth excluding priority facilities and infrastructures is 13.14% per year;
4. The growth in unit prices for classrooms, VPRs, teacher rooms, and toilets is 8.5% per year;
5. There is VPT information in the school profile data, but there are no rules for the number, type, and name of the ideal VPT for each skill and for each school and the unit price per VPT or skill. Therefore, the ideal number of VPTs is expressed as 0 units.

From this behavior and from running several scenarios, it can be concluded that the fulfillment of the priority infrastructure budget will be the same between the proposals and approved in 2026, while for infrastructures, there will be the same amount between the existing amount and the ideal amount in 2028. If you want to accelerate the fulfillment of infrastructures 2–3 years faster, then the provincial government of Central Kalimantan can carry out scenarios 2, 3, 4, and 5, namely allocating the infrastructure budget for only priority facilities and infrastructures, increasing the trend of the approved infrastructure budget doubling every year, allocating the infrastructure budget to fulfilling classrooms first, and using the rest for other priority infrastructures. However, the simulation results may take longer to fulfill if there is clear information on the vocational practice tools, both the current amount, the ideal amount, and the unit price of each practice tool for each skill.

**Author Contributions:** H.A.K.: conceptualization, data curation, formal analysis, funding, acquisition, methodology, project administration, and writing original draft; M.B.A.: supervision, validation, and review; Y.S.S.: supervision, validation, and review. All authors have read and agreed to the published version of the manuscript.

**Funding:** The APC was funded by Universitas Padjadjaran.

**Institutional Review Board Statement:** Not applicable.

**Informed Consent Statement:** Not Applicable.

**Data Availability Statement:** Data are contained within the article.

**Conflicts of Interest:** The authors declare no conflict of interest.

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
