# Peer review of "Analysis Projection of the Fulfillment of Priority Facilities and Infrastructures for Vocational High School/Sekolah Menengah Kejuruan (SMK) Using System Dynamic to Increase School Participation Rates in Central Kalimantan Province, Indonesia"

_sustainability, doi:10.3390/su152416696_

Round 1

Reviewer 1 Report

Comments and Suggestions for Authors

Dear Authors,

Your manuscript is an examination of the vocational education system in Indonesia, and the possible modelation of future expanses. The text need a major revision. 

Introduction

How is the history and the actual functionality of the vocational higher education system in Central Kalimantan? The text needs de description of the country and the presentation of the functionality, the present infrastructure of the education system. Complete with a graphic about the GDP of the country and the expenses for education system between 2017-2022. How is the youth poverty in this place? Complete with the general characteristics of the place, education system, schools and youths.

Rows 41-54 need reformulation. Must explain clearer!

Rows 48-54: What are the reasons for the differences in NER and GER? The text need explications.

Every abbreviation need explication. What it means?

How was the Budget Execution List (DPA) of the Education Office from 2017 to 2022? The text needs a graphic about these.

Materials and methods

It is very important the precise definition of study variables, the levels of measurement and the methods of your decisions.

Abstract: the results need reformulation, every abbreviation need explication.

References: rewrite concerning the MDPI requirements. 

With respect.

Author Response

For research article Analysis Projection of the Fulfillment of Priority Facilities and Infrastructure Vocational High School/Sekolah Menengah Kejuruan (SMK) Using System Dynamic to Increase School Participation Rates in Central Kalimantan Province, Indonesia

Response to Reviewer 1 Comments

1. Summary

Thank you very much for taking the time to review this manuscript. Please find the detailed responses below and the corresponding revisions in the re-submitted files.

2. Questions for General Evaluation

Reviewer’s Evaluation

Response and Revisions

Is the content succinctly described and contextualized with respect to previous and present theoretical background and empirical research (if applicable) on the topic?

Must be improved

Agree with the reviewer

Are all the cited references relevant to the research?

Can be improved

Agree with the reviewer

Is the research design, questions, hypothesis and methods clearly stated?

Must be improved

Agree with the reviewer

Are the arguments and discussion of findings coherent, balanced and compelling?

Must be improved

Agree with the reviewer

For empirical research, are the results clearly presented?

Must be improved

Agree with the reviewer

Is the article adequately referenced?

Can be improved

Agree with the reviewer

Are the conclusions thoroughly supported by the results presented in the article or referenced in secondary literature?

Can be improved

Agree with the reviewer

3. Point-by-point response to Comments and Suggestions for Authors

Comments 1: Your manuscript is an examination of the vocational education system in Indonesia, and the possible modelation of future expanses. The text need a major revision.

Response 1: Thank you for pointing this out. We agree with this comment. Therefore, we have changed the manuscript.

Comments 2: How is the history and the actual functionality of the vocational higher education system in Central Kalimantan? The text needs de description of the country and the presentation of the functionality, the present infrastructure of the education system. Complete with a graphic about the GDP of the country and the expenses for education system between 2017-2022. How is the youth poverty in this place? Complete with the general characteristics of the place, education system, schools and youths.

Response 2: Agree. We have, accordingly, changed to emphasize this point. The changement can be found at page number 2, paragraph 1, line 61 to 85 and page number 16, paragraph 1 and line 593 to 606. Therefore we have changed in the manuscript.

Comment 3: Rows 41-54 need reformulation. Must explain clearer!

Respons 3: In our humble opinion, the explanation is still giving clear information. We just change the APM and APK abbreviation to NPR (Net Participation Rate) and GPR (Gross Participation Rate) which is known widely. The changement can be found at page number 1, paragraph 1, line 35

Comment 4: Rows 48-54: What are the reasons for the differences in NER and GER? The text need explications

Response 4: The Abbreviation has changed. It is NPR (Net Participation Rate) and GPR (Gross Participation Rate) which is known widely. So we do not give explanation about NPR and GPR anymore.

Comment 5: Every abbreviation need explication. What it means?

Response 5: Agree. We have changed and explained every abbreviation in Indonesian term as follows:

1.        SMK = Sekolah Menengah Kejuruan or Vocational High School in English at page number 1, Title, line 4;

2.        DPA = Daftar Pelaksanaan Anggaran or Budget Execution List in English at page number 1, Abstract, line 16;

3.        SPR is School Participation Rate at page number 1, paragraph 1, line 31;

4.        NPR is Net Participation rate at page number 1, paragraph 1, line 35;

5.        GPR is Gross Participation Rate at page number 1, paragraph 1, line 35;

6.        APBN is Anggaran Pendapatan dan Belanja Negara or state budget in English at page number 1, paragraph 1, line 38;

7.        NPD = Neraca Pendidikan Daerah or Regional Education Balance in English at page number 1, paragraph 2, line 41;

8.        SMA = Sekolah Menengah Atas or Senior High School in English at page number 1, paragraph 1, line 44;

9.        SMP = Sekolah Menengah Pertama or Junior High School in English at page number 2, paragraph 1, line 55;

10.     USB = Unit Sekolah Baru or New School Units in English at page number 3, paragraph 2, line 94 to 95;

11.     BOS = Bantuan Operasional Sekolah or School Operasional Assistance in English at page number 4, paragraph 1, line 128 to 129;

12.     Iduka = Industri dan Dunia Kerja or Industry and the world of work in English at page number 4, paragraph 1, line 132 to 133;

13.     Dapodik = Data Pokok Pendidikan is the name of application for education database at page number 6, paragraph 1, line 232;

14.     PSMK Division is Pembinaan Sekolah Menengah Kejuruan or Vocational High School Management Division in English at page number 7 to 8, paragraph 4 and 1, line 330 to 331;

15.     DPRD = Dewan Perwakilan Rakyat Daerah or House of Local Representatives in English at page number 9, paragraph 7, line 401 to 402;

16.     PPAS = Prioritas Plafon Anggaran Sementara or Provisional Budget Ceiling Priority in English at page number 9, paragraph 7, line 402 to 403;

17.     KUA = Kebijakan Umum Anggaran or General Budget Policy in English at page number 9, paragraph 7, line 404.

Comment 6: How was the Budget Execution List (DPA) of the Education Office from 2017 to 2022? The text needs a graphic about these

Response 6: Agree. We have changed and explained including a table and a graphic about DPA 2017-2022 at page number 8, paragraph 1, line 334 to 335.

Comment 7: Materials and methods. It is very important the precise definition of study variables, the levels of measurement and the methods of your decisions.

Response 7: Agree. We have changed the explanation in the material and methods. I can be found at page number 6 to 8, line 276 to 354.

Comment 8: Abstract: the results need reformulation, every abbreviation need explication.

Response 8: Agree. We have changed the explanation in abstract. It can be found at page number 1, line 11 to 23.

Comment 9: References: rewrite concerning the MDPI requirements.

Response 9: Agree. We have deleted it

4. Response to Comments on the Quality of English Language

Point 1: I am not qualified to assess the quality of English in this paper

Response 1:    We have given a good explanation in English.

5. Additional clarifications

We hope, we have given clearly explanations in the manuscript.

Reviewer 2 Report

Comments and Suggestions for Authors

In the article, an analysis of the financial data regarding the infrastructure of the educational system in the Provincial Government of Central Kalimantan is proposed.

Authors should pay attention to following aspects:

1. There are very few bibliographic references, which indicates a very brief analysis of the specialized literature.

3. What are the research questions/hypotheses?

3. The research methodology is briefly elaborated.

4. In the discussion section, a correlation of the results of the current study with the data of other studies related to the infrastructure of the educational system is necessary.

5. The limits of the research are not highlighted.

Author Response

For research article Analysis Projection of the Fulfillment of Priority Facilities and Infrastructure Vocational High School/Sekolah Menengah Kejuruan (SMK) Using System Dynamic to Increase School Participation Rates in Central Kalimantan Province, Indonesia

Response to Reviewer 2 Comments

1. Summary

Thank you very much for taking the time to review this manuscript. Please find the detailed responses below and the corresponding revisions in the re-submitted files.

2. Questions for General Evaluation

Reviewer’s Evaluation

Response and Revisions

Is the content succinctly described and contextualized with respect to previous and present theoretical background and empirical research (if applicable) on the topic?

Must be improved

Agree with the reviewer

Are all the cited references relevant to the research?

Must be improved

Agree with the reviewer

Are the research design, questions, hypotheses and methods clearly stated?

Must be improved

Agree with the reviewer

Are the arguments and discussion of findings coherent, balanced and compelling?

Can be improved

Agree with the reviewer

For empirical research, are the results clearly presented?

Can be improved

Agree with the reviewer

Is the article adequately referenced?

Must be improved

Agree with the reviewer

Are the conclusions thoroughly supported by the results presented in the article or referenced in secondary literature?

Yes

Agree with the reviewer

3. Point-by-point response to Comments and Suggestions for Authors

Comments 1: In the article, an analysis of the financial data regarding the infrastructure of the educational system in the Provincial Government of Central Kalimantan is proposed.

Response 1: Thank you for pointing this out. We agree with this comment. Therefore, we have changed the manuscript. The analysis can be found at page number 8, table 1 and figure 2, at line 334 to 335;

Comments 2: Authors should pay attention to following aspects: 1. There are very few bibliographic references, which indicates a very brief analysis of the specialized literature.

Response 2: Agree. We have, accordingly, modified the manuscript to emphasize this point. In the manuscript, there are 49 bibliographic references.

Comment 3: 2. What are the research questions/hypotheses?

Response 3: We have changed in the manuscript. The research question is at the page number 6, paragraph 2, line 236 to 239. We have also given the hypoteses at page number 8, paragraph 4, line 348 to 354.

Comment 4: 3. The research methodology is briefly elaborated.

Response 4: We agree to the comment but we have not given any change. We think we have enough explained the research methodology about System Dynamic. There are many other explanations about system dynamic but we just give an explanation about the definition, benefit, and principles about it. Also, we write a little function of Vensim application.

Comment 5: 4. In the discussion section, a correlation of the results of the current study with the data of other studies related to the infrastructure of the educational system is necessary.

Response 5: Agree. We have changed the Discussion part and the explanation can be found at page number 15 to 17, line 573 to 652.

Comment 6: 5. The limits of the research are not highlighted.

Response 6: Agree. We have changed the manuscript and the explanation can be found at page number 6, paragraph 3 to 5, line 243 to 275.

4. Response to Comments on the Quality of English Language

Point 1: I am not qualified to assess the quality of English in this paper

Response 1:    We have given a good explanation in English.

5. Additional clarifications

We hope, we have given clearly explanations in the manuscript.

Reviewer 3 Report

Comments and Suggestions for Authors

The research uses a quantitative method in the form of a dynamic system 12 with the help of the vensim PLE64 application version 9.2.0.0 using profile data. The fact that the research is exclusively conducted within the framework of a certain institution is significant in assessing the effectiveness of the study and serves as a useful case study for identifying the guiding principles as well as the core values implemented.  

The references are well arranged, punctuation marks, word usage are quite fluid and correct in terms of language, but there seems to be a conflict between the title of the article and the content of the article.

The title "Analysis of Providing Priority Facilities and Infrastructure to Increase School Participation Rates" is carried out based on the argument that choice of education is only related to school conditions, and that to increase participation school environment can be provided by increasing the standards of physical features.

However, this argument is not completely valid for all countries and cultures mentioned in the article, and its realistic perspective cannot be considered correct. On the one hand, education might become trendy from time to time, which is linked to the value attributed to education, which in turn may be linked to other issues, such as Covid-19 restrictions or the value of agricultural products. These variables were not discussed in the text, yet mostly the number of facilities and infrastructure were considered.  When there is a growing intensity of participation, it might be figured out that investments in education could increase and the infrastructure is strengthened.

However, it seems logical to develop standards through Vensim in order to handle different geographies, countries, societies and values in a very different way and to create a policy model regarding the issue. Therefore, when considered through the presented scenarios, it is thought to be a very important study that can contribute to the field to a certain extent.

However, it is believed that it might not be enough for broad generalizations to be made based solely on one example since each institution and each different culture has its own potential, shortcomings, budget priority or system regarding the working principles. 

Author Response

For research article Analysis Projection of the Fulfillment of Priority Facilities and Infrastructure Vocational High School/Sekolah Menengah Kejuruan (SMK) Using System Dynamic to Increase School Participation Rates in Central Kalimantan Province, Indonesia

Response to Reviewer 3 Comments

1. Summary

Thank you very much for taking the time to review this manuscript. Please find the detailed responses below and the corresponding revisions in the re-submitted files.

2. Questions for General Evaluation

Reviewer’s Evaluation

Response and Revisions

Is the content succinctly described and contextualized with respect to previous and present theoretical background and empirical research (if applicable) on the topic?

Yes

Agree with the reviewer

Are all the cited references relevant to the research?

Yes

Agree with the reviewer

Are the research design, questions, hypotheses and methods clearly stated?

Yes

Agree with the reviewer

Are the arguments and discussion of findings coherent, balanced and compelling?

Yes

Agree with the reviewer

For empirical research, are the results clearly presented?

Yes

Agree with the reviewer

Is the article adequately referenced?

Yes

Agree with the reviewer

Are the conclusions thoroughly supported by the results presented in the article or referenced in secondary literature?

Can be improved

Agree with the reviewer

3. Point-by-point response to Comments and Suggestions for Authors

Comments 1: The research uses a quantitative method in the form of a dynamic system 12 with the help of the vensim PLE64 application version 9.2.0.0 using profile data. The fact that the research is exclusively conducted within the framework of a certain institution is significant in assessing the effectiveness of the study and serves as a useful case study for identifying the guiding principles as well as the core values implemented.

Response 1: Thank you for pointing this out. We agree with this comment.

Comments 2: The references are well arranged, punctuation marks, word usage are quite fluid and correct in terms of language, but there seems to be a conflict between the title of the article and the content of the article.

Response 2: Agree. We have, accordingly, revised the title of the article to emphasize this point to eliminate the conflict between the title of the article and the content of the article.

Comments 3: The title "Analysis of Providing Priority Facilities and Infrastructure to Increase School Participation Rates" is carried out based on the argument that choice of education is only related to school conditions, and that to increase participation school environment can be provided by increasing the standards of physical features.

Response 3: Agree with the reviewer.

Comments 4: However, this argument is not completely valid for all countries and cultures mentioned in the article, and its realistic perspective cannot be considered correct. On the one hand, education might become trendy from time to time, which is linked to the value attributed to education, which in turn may be linked to other issues, such as Covid-19 restrictions or the value of agricultural products. These variables were not discussed in the text, yet mostly the number of facilities and infrastructure were considered.  When there is a growing intensity of participation, it might be figured out that investments in education could increase and the infrastructure is strengthened.

Response 4: We agree to the comment but we just point the facilities and infrastructure and do not research about the other issues that can be linked to increase School Participation Rate even though they may be attributed. Regardless, we have changed the manuscript to minimize the ambiguity of explanation. The explanation can be found page number 6, paragraph 5, line 268 to 275.

Comments 5: However, it seems logical to develop standards through Vensim in order to handle different geographies, countries, societies and values in a very different way and to create a policy model regarding the issue. Therefore, when considered through the presented scenarios, it is thought to be a very important study that can contribute to the field to a certain extent.

Response 5: Agree. Vensim is the right application to simulate and make the projection of fulfillment of SMK facilities and infrastructure.

Comments 6: However, it is believed that it might not be enough for broad generalizations to be made based solely on one example since each institution and each different culture has its own potential, shortcomings, budget priority or system regarding the working principles.

Response 6: We agree and disagree to the comment. With the same method and datas, it might be implemented to the condition in the other province in Indonesia but it might not be implemented to the condition in the other country because of the diffenrence of budget policy and the education information system application.

4. Response to Comments on the Quality of English Language

Point 1: English language fine. No issues detected

Response 1:    Agree

5. Additional clarifications

We hope, we have given clearly explanations in the manuscript.

Reviewer 4 Report

Comments and Suggestions for Authors

The article entitled: Analysis of the Fulfillment of Priority Facilities and Infrastructure to Increase School Participation Rates is a correct proposal framed in the cultural and economic dimensions that concern the journal Sustainability. Undoubtedly, school participation and the conditions and infrastructures that are linked to it is a relevant topic for both the social and educational spheres, as well as for sustainable development.

The structure of the contribution corresponds to the requirements of the scientific community, respecting a theoretical introduction and a concrete method based on data collection and analysis using a statistical programme (Vensim). The creation of the constructs (also called variables by the authors) is appropriate.

In general, the article is interesting and coherent and can be accepted by the journal. For this, from the evaluator's point of view, they should modify/add:

- Give the text a more international character. That is, it is focused on the Borneo islands and this cannot be changed, but it could be compared with other areas of the world and explain what global benefits this study can have.

- Include more references related to education policy and the investments made in it. Along the lines of the reference already included in the text: B. Högberg, "Transitions from Unemployment to Education in Europe: The Role of Educational Policies," J. Soc. Policy, vol. 618 48, no. 4, pp. 699-720, 2019, doi: 10.1017/S0047279418000788. In other words, increase the number of references by orienting them towards the topic in question.

- Increase the number of references that involve articles or book chapters, replacing or complementing those that refer to speeches or documentation.

Author Response

For research article Analysis Projection of the Fulfillment of Priority Facilities and Infrastructure Vocational High School/Sekolah Menengah Kejuruan (SMK) Using System Dynamic to Increase School Participation Rates in Central Kalimantan Province, Indonesia

Response to Reviewer 4 Comments

1. Summary

Thank you very much for taking the time to review this manuscript. Please find the detailed responses below and the corresponding revisions in the re-submitted files.

2. Questions for General Evaluation

Reviewer’s Evaluation

Response and Revisions

Is the content succinctly described and contextualized with respect to previous and present theoretical background and empirical research (if applicable) on the topic?

Must be improved

Agree with the reviewer

Are all the cited references relevant to the research?

Must be improved

Agree with the reviewer

Are the research design, questions, hypotheses and methods clearly stated?

Yes

Agree with the reviewer

Are the arguments and discussion of findings coherent, balanced and compelling?

Yes

Agree with the reviewer

For empirical research, are the results clearly presented?

Yes

Agree with the reviewer

Are the conclusions supported by the results?

Can be improved

Agree with the reviewer

Are the conclusions thoroughly supported by the results presented in the article or referenced in secondary literature?

Yes

Agree with the reviewer

3. Point-by-point response to Comments and Suggestions for Authors

Comments 1: The article entitled: Analysis of the Fulfillment of Priority Facilities and Infrastructure to Increase School Participation Rates is a correct proposal framed in the cultural and economic dimensions that concern the journal Sustainability. Undoubtedly, school participation and the conditions and infrastructures that are linked to it is a relevant topic for both the social and educational spheres, as well as for sustainable development.

Response 1: Thank you for pointing this out. We agree with this comment

Comments 2: The structure of the contribution corresponds to the requirements of the scientific community, respecting a theoretical introduction and a concrete method based on data collection and analysis using a statistical programme (Vensim). The creation of the constructs (also called variables by the authors) is appropriate.

Response 2: Thank you for pointing this out. We agree with this comment

Comments 3: In general, the article is interesting and coherent and can be accepted by the journal. For this, from the evaluator's point of view, they should modify/add:

- Give the text a more international character. That is, it is focused on the Borneo islands and this cannot be changed, but it could be compared with other areas of the world and explain what global benefits this study can have.

Response 3: Agree. We have changed in the manuscript. Many abbreviations can be changed to international abbreviation because of characteristic of Central Kalimantan Province and Indonesia and also the name of the document but the researcher tries to provide terms in English. Here is the abbreviation that in researcer opinion can not be changed and what is the term in English:

1.       SMK = Sekolah Menengah Kejuruan or Vocational High School in English at page number 1, Title, line 4;

2.       DPA = Daftar Pelaksanaan Anggaran or Budget Execution List in English at page number 1, Abstract, line 16;

3.       APBN is Anggaran Pendapatan dan Belanja Negara or state budget in English at page number 1, paragraph 1, line 38;

4.       NPD = Neraca Pendidikan Daerah or Regional Education Balance in English at page number 1, paragraph 2, line 41;

5.       SMA = Sekolah Menengah Atas or Senior High School in English at page number 1, paragraph 2, line 44;

6.       SMP = Sekolah Menengah Pertama or Junior High School in English at page number 2, paragraph 1, line 55;

7.       USB = Unit Sekolah Baru or New School Units in English at page number 3, paragraph 2, line 94 to 95;

8.       BOS = Bantuan Operasional Sekolah or School Operasional Assistance in English at page number 4, paragraph 1, line 128 to 129;

9.       Iduka = Industri dan Dunia Kerja or Industry and the world of work in English at page number 4, paragraph 1, line 132 to 133;

10.    Dapodik = Data Pokok Pendidikan is the name of application for education database at page number 6, paragraph 1, line 232;

11.    PSMK Division is Pembinaan Sekolah Menengah Kejuruan or Vocational High School Management Division in English at page number 7 to 8, paragraph 4 and 1, line 330 to 331;

12.    DPRD = Dewan Perwakilan Rakyat Daerah of House of Local Representatives in English at page number 9, paragraph 7, line 401 to 402;

13.    PPAS = Prioritas Plafon Anggaran Sementara or Provisional Budget Ceiling Priority in English at page number 9, paragraph 7, line 402 to 402;

14.    KUA = Kebijakan Umum Anggaran or General Budget Policy in English at page number 9, paragraph 7, line 404.

Comments 4: - Include more references related to education policy and the investments made in it. Along the lines of the reference already included in the text: B. Högberg, "Transitions from Unemployment to Education in Europe: The Role of Educational Policies," J. Soc. Policy, vol. 618 48, no. 4, pp. 699-720, 2019, doi: 10.1017/S0047279418000788. In other words, increase the number of references by orienting them towards the topic in question.

Response 4: Agree. We have a maximum effort for this and now, there are 49 references in total.

Comments 5: - Increase the number of references that involve articles or book chapters, replacing or complementing those that refer to speeches or documentation.

Response 5: Agree. We have a maximum effort for this and now, there are 49 references in total.

4. Response to Comments on the Quality of English Language

Point 1: I am not qualified to assess the quality of English in this paper

Response 1:    We have given a good explanation in English

5. Additional clarifications

We hope, we have given clearly explanations in the manuscript.

Round 2

Reviewer 1 Report

Comments and Suggestions for Authors

In my view, it is ok. 

Author Response

For research article

Response to Reviewer 1 Comments

1. Summary

2. Questions for General Evaluation

Reviewer’s Evaluation

Response and Revisions

Is the content succinctly described and contextualized with respect to previous and present theoretical background and empirical research (if applicable) on the topic?

Can be improved

We agree with reviewer. We tryed to find previous research about projection using system dynamic, but I did not find it

Are all the cited references relevant to the research?

Can be improved

We agree with reviewer. We have added 11 new referencies so there are 60 referencies now. Those are 1) reference no. 20, page number 3, paragraph 2, line 106 to 109; 2) reference no. 33, page number 4, paragraph 4, line 172 to 174; 3) reference no. 44, page number 17, paragraph 3, line 638; 4) reference no. 51 and 52, page number 17, paragraph 3, line 645 to 651; 5) reference no. 55, 56, 57, 58, 59, and 60, page number 18, paragraph 2, line 666 to 692.

Are the research design, questions, hypotheses and methods clearly stated?

Can be improved

We agree with reviewer. But in our humble opinion, this is the best describe we can provide.

Are the arguments and discussion of findings coherent, balanced and compelling?

Can be improved

We agree with the reviewer. We have added our arguments and discussion as shown in green highlight in the manuscript: 1) page number 16, paragraph 3, line 603 to 604;    

For empirical research, are the results clearly presented?

Can be improved

We agree with the reviewer. We have added 1 results in the manuscript at poin 3.3.5. page number 15, paragraph 2, line 570 to page number 16, paragraph 2, line 598.

Is the article adequately referenced?

Can be improved

We agree with the reviewer. We have added 11 new referencies so there are 60 referencies now. Those are 1) reference no. 20, page number 3, paragraph 2, line 106 to 109; 2) reference no. 33, page number 4, paragraph 4, line 172 to 174; 3) reference no. 44, page number 17, paragraph 3, line 638; 4) reference no. 51 and 52, page number 17, paragraph 3, line 645 to 651; 5) reference no. 55, 56, 57, 58, 59, and 60, page number 18, paragraph 2, line 666 to 692.

Are the conclusions thoroughly supported by the results presented in the article or referenced in secondary literature?

Can be improved

We agree with the reviewer. But in our humble opinion, this is the best describe we can provide.. In additon, we add scenario 5 at page 19, paragraph 2, line 732

3. Point-by-point response to Comments and Suggestions for Authors

Comments 1: In my view, it is ok.  

Response 1: Thank you for pointing this out. We agree with this comment. Therefore, we have added the manuscript as shown in green highlight that we have described at point 2 Response and Revision

4. Response to Comments on the Quality of English Language

Point 1: I am not qualified to assess the quality of English in this paper

Response 1: We have given a good explanation in English as good as possible.

5. Additional clarifications

We hope, we have given clearly explanation in the manuscript
